# Barriers and Facilitators to Proactive Deprescribing in Saudi Hospitals: A Qualitative Study Using the Theoretical Domains Framework

**DOI:** 10.3390/healthcare13111274

**Published:** 2025-05-28

**Authors:** Mohammed S. Alharthi

**Affiliations:** Department of Clinical Pharmacy, College of Pharmacy, Taif University, Taif 21944, Saudi Arabia; ms.harthi@tu.edu.sa

**Keywords:** proactive deprescribing, polypharmacy, barriers, facilitators, Theoretical Domains Framework

## Abstract

Background: Polypharmacy, commonly defined as the use of five or more medications, is a growing concern in hospitals due to its association with adverse drug reactions, functional decline, and increased healthcare costs. Proactive deprescribing, which involves the planned discontinuation of unnecessary or potentially harmful medications, can optimise medication use. However, multiple barriers hinder its implementation. Saudi Arabia offers a unique context for deprescribing due to strong family roles in care, prevalent prescribing norms, and ongoing shifts toward value-based healthcare. This study explores the barriers and facilitators to proactive deprescribing among physicians in Saudi hospitals using the Theoretical Domains Framework (TDF). The TDF was used as it effectively identifies behavioural factors influencing clinical decision making in practice. Methods: Semi-structured interviews were conducted with 27 purposively sampled physicians experienced in managing polypharmacy. The interviews were transcribed and analysed thematically, with behavioural determinants identified and categorised according to the 14 domains of the Theory of Planned Behaviour (TDF). Results: Enablers included the availability of deprescribing guidelines, decision–support tools, interprofessional collaboration, and institutional backing. Physicians with specialised training expressed greater confidence in conducting deprescribing. Identified barriers included limited time, heavy workload, absence of standardised protocols, medico-legal concerns, resistance from patients and caregivers, and lack of formal training. These factors were categorised under seven key TDF domains, with Environmental Context and Resources, Social Influences, and Beliefs About Capabilities identified as the most influential in shaping physicians’ deprescribing practices. Interactions between factors were observed, where supportive environments and collaborative teams helped offset key barriers such as time constraints, legal concerns, and patient resistance. Conclusions: This study identified key behavioural and contextual factors influencing proactive deprescribing in Saudi hospital settings. Addressing barriers such as heavy workload, medico-legal concerns, and lack of standardised protocols through targeted interventions, including clinician training, institutional support, and multidisciplinary collaboration, may facilitate the integration of deprescribing into routine practice. The findings offer context-specific insights to inform future efforts aimed at improving medication safety and optimising prescribing in the Saudi healthcare system.

## 1. Introduction

Polypharmacy, commonly defined as the concurrent use of five or more medications, is increasingly prevalent due to the growing multimorbidity in ageing populations [1]. While polypharmacy may be necessary for managing multiple chronic conditions, it is associated with significant risks, including adverse drug reactions (ADRs), increased hospitalisations, functional decline, and medication non-adherence [2]. Inappropriate prescribing exacerbates these risks and contributes to a higher healthcare burden [3]. Studies have shown that the higher the number of medications, the greater the risk of drug–drug interactions, treatment burden, and compromised patient adherence, leading to suboptimal therapeutic outcomes [4,5]. Deprescribing has emerged as a critical strategy to mitigate the risks of polypharmacy [6]. Deprescribing is a systematic, patient-centred process of reducing or stopping medications that may no longer be beneficial or may cause harm [7]. The literature underscores deprescribing as an evidence-based intervention that improves medication safety, reduces ADRs, and enhances patient well-being [8,9,10]. However, the deprescribing process is complex and influenced by multiple factors, including clinician perceptions, patient resistance, and systemic barriers [11]. The reluctance to deprescribe often stems from physician uncertainty about the long-term consequences of stopping medications, lack of training in deprescribing, and fear of patient dissatisfaction, particularly when medications have been prescribed for extended periods [12,13]. A critical distinction exists between reactive and proactive deprescribing. Reactive deprescribing occurs in response to an adverse drug event (ADE) or new symptoms attributed to medications. It is an essential but insufficient approach, as it fails to prevent medication-related harm before it occurs. Proactive deprescribing, on the other hand, involves routinely evaluating a patient’s medication regimen to prevent potential harm before symptoms [14,15]. This approach is embedded in medication therapy management and structured medication reviews, with guidelines advocating periodic reassessment of polypharmacy, particularly in vulnerable populations [16]. The estimated prevalence of polypharmacy varies widely, with recent studies reporting rates as high as 40–60% among older adults, depending on the definition and method used [17]. Collaborative efforts with clinical pharmacists have been shown to significantly reduce polypharmacy and inappropriate prescribing, particularly during transitions of care [18]. Research has demonstrated that proactive deprescribing improves patient outcomes, including enhanced quality of life, reduced hospital readmissions, and minimised medication-related complications [19]. Despite its recognised benefits, proactive deprescribing is not widely implemented due to multiple barriers. Physicians often cite lack of time, uncertainty about deprescribing protocols, concerns about potential withdrawal effects, and medico-legal implications as deterrents [20]. In the hospital setting, the transient nature of patient encounters, fragmented communication between care providers, and competing clinical priorities hinder proactive deprescribing efforts [21]. There is also a gap in research exploring physicians’ perspectives on deprescribing within hospitals, particularly in low- and middle-income countries where healthcare systems may lack structured deprescribing protocols. Physicians often feel pressured to adhere to traditional prescribing patterns, and deprescribing is sometimes perceived as contrary to established medical practices [22,23]. In Saudi Arabia, physicians may face unique cultural and systemic barriers to deprescribing. Strong family involvement in care decisions can lead to resistance when medications are withdrawn, as caregivers often equate prescribing with better care [20,21]. Additionally, prescribing norms in some settings favour action over caution, making deprescribing seem counterintuitive [10]. Concerns about medico-legal liability further discourage physicians from proactively discontinuing medications, especially in complex or high-risk cases [18]. In Saudi hospital settings, reactive deprescribing commonly follows acute incidents such as falls, delirium, or renal impairment, triggering urgent medication reviews [10]. For instance, sedatives or anticholinergics may be discontinued only after adverse events occur. In contrast, proactive deprescribing—such as tapering unnecessary proton pump inhibitors or antihypertensives during routine medication reviews—is limited and typically observed in institutions with geriatric teams or pharmacist-led interventions [18]. These examples highlight the inconsistent integration of proactive deprescribing into routine hospital practice in Saudi Arabia. Given these challenges, understanding physicians’ perspectives on deprescribing in Saudi Arabian hospitals is crucial, particularly as the healthcare system evolves towards value-based care. Hospitals serve as critical environments for medication review, particularly during patient transitions between acute and primary care [24]. Identifying and addressing hospital-specific factors that facilitate or hinder proactive deprescribing can help develop targeted interventions to optimise medication use and reduce medication-related harm. Furthermore, addressing these issues within Saudi Arabia’s hospital settings is particularly significant, as the country faces a growing burden of chronic diseases, increased medication use, and a shifting healthcare landscape toward value-based care [25]. To systematically explore the behavioural factors influencing deprescribing, this study employed the Theoretical Domains Framework (TDF), a comprehensive model widely used in implementation research to identify determinants of healthcare professional behaviour and guide intervention development. TDF is a comprehensive model that consolidates multiple behaviour change theories to identify determinants of clinical behaviours [26]. This study aimed to explore the key barriers and facilitators influencing physicians’ engagement with proactive deprescribing in hospital settings in Saudi Arabia using the Theoretical Domains Framework (TDF). By identifying the main behavioural determinants, the study seeks to inform the development of targeted strategies to support routine deprescribing practices and enhance medication safety [20].

## 2. Methods

### 2.1. Study Design

This qualitative study employed the Theoretical Domains Framework (TDF) to examine the barriers and facilitators influencing physicians’ engagement with proactive deprescribing practices. TDF was utilised to identify and comprehend the specific challenges and enabling factors encountered by physicians when making proactive deprescribing decisions in hospitals.

### 2.2. Recruitment

Physicians were recruited from hospitals across various regions of Saudi Arabia to ensure diverse representation from different specialities. To ensure diversity in medical specialities, hospital types, and geographic regions across Saudi Arabia, potential participants were contacted via telephone or email, and informed written consent was obtained before participation. All participants provided informed written consent, and the study adhered to ethical standards approved by the institutional review board. The inclusion criteria required participants to have experience managing patients with polypharmacy and working in a hospital setting. There were no minimum experience requirements. Exclusion criteria encompassed physicians without experience in proactive deprescribing or those not directly involved in medication adjustment.

### 2.3. Ethical Approval

The Ethics Committee of Taif University (Application Number 46-145) granted ethical approval for the study, ensuring compliance with ethical research standards, including participant confidentiality and data protection throughout the study.

### 2.4. Sampling and Sample Size

A purposive sampling strategy was employed to recruit physicians from various medical specialities. Following the guidelines set by Francis et al. [27], an initial sample of 10 interviews was conducted, with additional interviews performed until data saturation was reached, ultimately resulting in 27 interviews. This approach allowed for a thorough exploration of diverse experiences and perspectives. The final sample size of 27 physicians was sufficient to achieve thematic saturation, with no new themes emerging after the 24th interview.

### 2.5. Interview Procedure and Data Collection

Interview questions were developed from a literature review and consultations with two experienced academics [28,29,30]. The researcher, trained in qualitative interviewing techniques, conducted and audio-recorded the interviews, which were transcribed verbatim. At the start of each interview, participants provided demographic and professional details, including age, gender, years of experience, and speciality. Interviews ranged from 30 min to one hour and were guided by open-ended questions exploring the barriers and facilitators of proactive deprescribing practices. Semi-structured interviews minimised bias, allowing for consistent data collection while providing flexibility for an in-depth exploration of physicians’ experiences. To minimise researcher bias, reflexivity was maintained through memo writing and peer debriefing during data analysis. All interviews were conducted by the lead researcher, trained in qualitative interviewing.

### 2.6. Data Analysis Coding Process

An interim analysis was performed following the first 10 interviews, with coding continuing until data saturation was achieved at 27 interviews. Participant characteristics were summarised, and the transcripts were manually coded according to the 14 TDF domains [26]. Each coded excerpt was synthesised into belief statements and then categorised under the relevant TDF domains. The researcher kept memos and an audit trail to enhance rigour and trustworthiness, using verbatim quotes to support belief statements. The Theoretical Domains Framework (TDF) is an integrative model that synthesises multiple behaviour change theories and is widely employed in healthcare research to identify and understand the factors influencing behaviour, thereby informing intervention development. The TDF consists of 14 domains, including Knowledge, Skills, Social/Professional Role and Identity, Beliefs about Capabilities, Optimism, Beliefs about Consequences, Reinforcement, Intentions, Goals, Memory, Attention and Decision Processes, Environmental Context and Resources, Social Influences, Emotion, and Behavioural Regulation. The TDF was selected for this study due to its alignment with behaviour change techniques (BCTs), which are key components of interventions, facilitating the identification of practical elements for designing theory-based interventions to modify practitioner behaviour. Furthermore, the TDF encompasses 128 theoretical constructs derived from 33 behaviour change theories, grouped into 14 domains [27]. Using the TDF, this study identified the multifaceted barriers and enablers influencing physicians’ engagement with proactive deprescribing practices. After transcription, the interview data were read multiple times to ensure familiarisation. Segments of text reflecting behavioural influences were manually coded and mapped to the 14 TDF domains. A deductive thematic analysis was conducted, with coding structured around the 14 domains of the Theoretical Domains Framework (TDF). Belief statements were then generated from coded excerpts and grouped into higher-order themes representing key barriers and facilitators. This thematic coding process was iterative and supported by memo writing and peer debriefing to ensure analytical rigour and consistency. To ensure transparent reporting, the study was reviewed and documented following the Consolidated Criteria for Reporting Qualitative Research (COREQ) checklist (Appendix A).

### 2.7. Trustworthiness

To ensure trustworthiness, several strategies were employed. Credibility was supported through purposive sampling, peer debriefing with experienced qualitative researchers, and the use of verbatim quotes. Dependability and confirmability were enhanced by maintaining an audit trail, engaging in reflexive journaling, and documenting analytical decisions throughout the coding process. These steps strengthened the rigour and transparency of the study.

## 3. Results

Table 1 shows that the study included a diverse group of physicians with varying demographic and professional backgrounds. Most participants were male (22; 81.5%), while female physicians made up 5 participants (18.5%). The age distribution spanned different career stages, with significant representation from mid-career and senior physicians. Thirteen participants (48.1%) had more than 20 years of experience, while eleven (40.7%) had between 11 and 20 years, highlighting the expertise of respondents in clinical decision making. In terms of specialisation, participants came from a wide range of medical fields, including paediatrics (6; 22.2%), neurology (6; 22.2%), internal medicine (3; 11.1%), general practice (2; 7.4%), and rehabilitation medicine (2; 7.4%), among others. Geographically, participants were distributed across different regions of Saudi Arabia. The Western region accounted for 8 participants (29.6%), including those practising in Jeddah, Makkah, and Taif. The Central region had the highest representation, with 9 participants (33.3%) from Al-Kharj, Riyadh, and Buraydah. The Southern region comprised 4 participants (14.8%), representing Abha and Jizan, while the Eastern region accounted for 6 participants (22.2%), including Dammam, Al-Khobar, and Jubail.

### 3.1. Enablers and Barriers to Proactive Deprescribing

The study identified seven key domains influencing physicians’ engagement in proactive deprescribing. These domains encompass various enablers and barriers that shape deprescribing practices. The identified domains are as follows:Environmental Context and Resources.Social Influences.Beliefs About Capabilities.Social/Professional Role and Identity.Skills.Beliefs About Consequences.Knowledge.

### 3.2. Environmental Context and Resources

Institutional resources and policies are crucial for proactive deprescribing, providing physicians with structured tools, deprescribing algorithms, and clinical pathways that facilitate informed decision making. These resources reduce uncertainty and support evidence-based discontinuation of unnecessary medications [31]. Institutions that integrate deprescribing into routine workflows foster a systematic approach to medication review, prioritising patient safety and quality care. Adequate consultation time is a key enabler, allowing physicians to assess medication regimens thoroughly and discuss deprescribing with patients and caregivers. Additionally, Electronic Health Records (EHRs) with deprescribing alerts and support from clinical pharmacists and nurses streamline workflows and facilitate multidisciplinary deprescribing efforts.

“Deprescribing tools make the process safer and more efficient. They give me confidence by providing strong evidence to support my decisions.”
**PD4.**


“Our institution supports deprescribing with policies and weekly case reviews, making identifying and removing unnecessary medications easier.” 
**PD7.**


Despite these facilitators, high patient volumes and limited consultation time often force physicians to prioritise urgent medical concerns over anticipatory deprescribing. The absence of standardised deprescribing guidelines tailored to local populations increases variability and reduces confidence in deprescribing decisions. Outdated EHR systems without deprescribing alerts further hinder proactive medication reviews. At the same time, a shortage of clinical pharmacists and support staff places the entire burden on physicians, increasing stress and delaying deprescribing interventions.

“Time constraints make proactive deprescribing difficult. We often focus on immediate clinical issues, and deprescribing gets pushed aside.” 
**PD15.**


“Our outdated hospital system makes medication reviews difficult. Without a centralised tool to flag inappropriate medications, I have to check each list manually, which is too time consuming.” 
**PD24.**


While institutional tools and policies supported deprescribing, high workload and system limitations often overrode these facilitators in daily practice.

### 3.3. Social Influences

Collaboration within multidisciplinary teams plays a crucial role in enabling proactive deprescribing [8]. Physicians emphasised that working alongside clinical pharmacists, geriatricians, and other specialists fosters a shared decision-making approach, allowing complex cases to be discussed and deprescribing decisions to be validated. Regular case discussions and multidisciplinary meetings provide structured platforms to exchange insights and refine deprescribing strategies. Peer support and mentorship further empower physicians, boosting their confidence in deprescribing and reducing hesitation. A workplace culture that prioritises team-based deprescribing helps distribute responsibility, making proactive deprescribing more manageable and effective.

“Discussions with clinical pharmacists and geriatricians help refine my deprescribing decisions and boost my confidence in the process.” 
**PD3.**


“Mentorship from senior colleagues gives me confidence in complex deprescribing cases. Early on, I hesitated, fearing mistakes, but their guidance helped me make informed decisions.” 
**PD9.**


Despite the benefits of collaboration, resistance from patients and caregivers often hinders proactive deprescribing efforts. Many patients perceive deprescribing as a reduction in care rather than a preventive measure, particularly if they have been on medication for an extended period. Caregivers, especially those deeply involved in a patient’s treatment, may express concerns about treatment withdrawal and advocate for continued medication use. Additionally, cultural and communication barriers, including language differences and low health literacy, make it challenging for physicians to convey the importance of proactive deprescribing. A lack of peer or supervisor support can further discourage physicians from taking initiative, particularly when deprescribing is perceived as a risky or unsupported clinical decision.

“Patients often resist deprescribing because they associate stopping a medication with worsening health. It takes time to convince them it’s preventive.” 
**PD6.**


“Caregivers often question my decisions, making it harder to proceed with deprescribing. They worry that stopping medication means giving up on treatment.” 
**PD10.**


Although teamwork enhanced confidence, patient and caregiver resistance frequently complicated shared decision-making efforts.

### 3.4. Beliefs About Capabilities

Confidence in their abilities was a recurring theme among physicians who actively deprescribed. Specialised training programs, such as workshops on deprescribing frameworks or medication review strategies, were identified as critical enablers. These programs equipped physicians with the practical skills and knowledge to address polypharmacy and its associated risks effectively. Experienced physicians managing complex cases reported greater competence and confidence in deprescribing, attributing it to their accumulated expertise. Continuous professional development opportunities also reinforced physicians’ beliefs in their capabilities. Access to up-to-date guidelines, research articles, and case studies gave them the knowledge to make informed decisions. Positive patient outcomes further bolstered this confidence, validating their deprescribing decisions and encouraging them to continue these practices.

“Training boosted my confidence in managing polypharmacy and deprescribing. I was hesitant before, but a workshop on deprescribing strategies reassured my decisions.” 
**PD2.**


“Handling complex cases over time has strengthened my deprescribing skills. I often doubted myself initially, but seeing positive outcomes has greatly boosted my confidence.” 
**PD9.**


“Seeing positive outcomes in patients strengthens my belief in the importance of deprescribing. When I deprescribe appropriately, and the patient’s health improves, it reinforces that I made the right call.” 
**PD16.**


Confidence gained from training was sometimes undermined by clinical uncertainty in complex deprescribing scenarios.

### 3.5. Social/Professional Role and Identity

A strong sense of professional responsibility and identity is crucial in enabling proactive deprescribing. Physicians who see deprescribing as part of their duty to provide high-quality, patient-centred care are likelier to engage in it. Many physicians reported feeling a moral obligation to review and optimise medication regimens, ensuring patients are not exposed to unnecessary medication risks. This perspective is reinforced by professional training, which emphasises evidence-based practice and patient safety. Moreover, physicians working in healthcare settings where deprescribing is integrated into routine practice felt more empowered to deprescribe. Institutions that promote medication optimisation as part of their organisational culture help reinforce deprescribing as a professional standard rather than an optional practice. Supportive colleagues and mentors further enhance physicians’ confidence in deprescribing, as they provide reassurance that deprescribing is a valued and encouraged aspect of patient care.

“As a physician, my role is to prescribe and reassess whether medications are still necessary. I must ensure my patients are not taking medications that might do them more harm than good.” 
**PD1.**


“I see deprescribing as an essential part of patient-centred care. It’s not about withholding treatment; it’s about making sure that every medication the patient takes is helping them.” 
**PD5.**


Conversely, physicians working in environments where deprescribing is not actively encouraged face significant challenges. A lack of institutional emphasis on deprescribing can lead to uncertainty or hesitation among physicians, as they may feel that deprescribing is not aligned with their professional role. In some cases, physicians reported that deprescribing was perceived as “going against the norm”, particularly in settings where prescribing more medications was seen as a sign of thorough care. Another barrier is patients’ and caregivers’ expectation that physicians should always prescribe something. Many patients associate prescriptions with high-quality care, and when a physician suggests stopping a medication, it may be misinterpreted as neglect or a lack of concern. This pressure can discourage physicians from deprescribing, even when they believe it is in the patient’s best interest.

“There is an unspoken culture in some settings where prescribing is seen as ‘doing more’ for the patient, and deprescribing is seen as taking something away. This makes it difficult to justify deprescribing, even when it’s the right decision.” 
**PD8.**


“Patients often expect me to prescribe something during every visit. When I suggest stopping a medication instead, they sometimes feel like they are not receiving proper care, which makes deprescribing more challenging.” 
**PD11.**


Physicians expressed a strong sense of professional responsibility; however, cultural expectations for continued prescribing at times conflicted with this.

### 3.6. Skills

Physicians with advanced training in medication management, geriatrics, or pharmacology reported greater confidence in deprescribing. Training programs, workshops, and hands-on experience significantly enhance physicians’ ability to assess medication necessity, understand potential withdrawal effects, and communicate deprescribing decisions effectively to patients. Those with strong critical thinking and decision-making skills are better equipped to navigate complex cases where deprescribing is not straightforward. Additionally, physicians with strong communication skills find it easier to discuss deprescribing with patients and caregivers. Clear explanations, reassurance, and shared decision-making approaches help build trust and make deprescribing smoother. Physicians also emphasised that experience over time helps refine their deprescribing skills, making them more efficient and confident in their decisions.

“Effective communication skills make a huge difference. When I explain deprescribing clearly and involve the patient in the decision-making process, they are much more likely to accept it.” 
**PD13.**


“Skill development over time has helped me deprescribe more effectively. I’ve learned to anticipate potential withdrawal effects and taper medications properly, which minimises risks.” 
**PD25.**


A significant barrier to deprescribing is the lack of formal training in medication discontinuation. Many physicians reported that while medical education extensively covers how to prescribe medications, there is limited training on when and how to stop them safely. This gap in training can lead to uncertainty, hesitation, and fear of unintended consequences, discouraging physicians from deprescribing. Another challenge is the variability in individual physicians’ skills and confidence levels. Physicians with limited exposure to deprescribing during their training may feel unequipped to handle complex cases involving multiple medications. Some also struggle with effective patient communication, finding it challenging to explain deprescribing decisions in a way that reassures patients and prevents resistance.

“I sometimes hesitate to deprescribe because I don’t feel fully confident about how to do it safely. Without proper training, I worry about making the wrong decision and causing harm.” 
**PD16.**


“Convincing patients to deprescribe is a skill in itself. If you don’t communicate well, they’ll refuse, and you’ll end up back at square one.” 
**PD23.**


Skills development empowered deprescribing, but gaps in formal training and communication skills remained significant obstacles.

### 3.7. Beliefs About Consequences

Physicians who have witnessed positive patient outcomes following deprescribing are more likely to engage in proactive medication reviews. Seeing patients experience fewer side effects, improved cognitive function, or better overall health after discontinuing unnecessary medications reinforces the belief that deprescribing is beneficial. Physicians also highlighted that deprescribing leads to better adherence to necessary medications since patients are no longer overwhelmed by complex regimens. Another enabling factor is the understanding that deprescribing aligns with best practices in patient safety. Physicians who are well-informed about the risks of polypharmacy and medication overuse recognise deprescribing as enhancing patient care rather than reducing it. This belief strengthens their commitment to deprescribing as an essential part of their medical practice.

“I’ve had patients who, after deprescribing, had fewer falls and better cognitive function. Seeing those improvements firsthand makes me even more committed to deprescribing.” 
**PD2.**


“Deprescribing isn’t about withholding care; it’s about refining care. I remind myself that stopping unnecessary medications often means better health outcomes for the patient.” 
**PD15.**


One of the main barriers is the fear of negative consequences, such as withdrawal effects, symptom recurrence, or potential legal liabilities. Physicians expressed concerns about deprescribing medications, particularly for older patients or those with chronic conditions, due to the possibility of adverse effects. This uncertainty often leads to a preference for maintaining the status quo rather than deprescribing. Another significant barrier is the perception that deprescribing might damage the physician–patient relationship. Physicians worry that patients may lose trust in their care if they feel their treatment is being “taken away.” This is particularly concerning for physicians who have longstanding relationships with their patients and want to maintain a sense of trust and continuity.

“I’ve seen cases where stopping a medication led to withdrawal symptoms, which made me hesitant to deprescribe in similar situations. I don’t want to risk causing harm.” 
**PD4.**


“I don’t want to damage the relationship with my patients. If they think I’m taking away their treatment, they might not trust me as much, and that’s a real concern.” 
**PD17.**


While positive patient outcomes reinforced deprescribing, fear of adverse effects and legal liability deterred some physicians from acting.

### 3.8. Knowledge

Physicians emphasised that access to up-to-date evidence, deprescribing guidelines, and structured educational resources support deprescribing efforts. They highlighted that well-defined guidelines provide clarity and reassurance when deprescribing decisions, reducing hesitation and uncertainty. Additionally, participation in deprescribing workshops and the availability of clinical decision–support tools were valuable in enhancing physicians’ confidence and competence in safely discontinuing medications. Some physicians noted that having institutional support and formal deprescribing protocols further reinforced their ability to engage in deprescribing practices.

“Workshops and training programs have helped me understand when and how to deprescribe. Without them, I wouldn’t feel as comfortable stopping medications, even when I know it’s necessary.” 
**PD16.**


“When deprescribing is part of institutional policies, it becomes a shared responsibility rather than just an individual decision, which makes it easier to implement.” 
**PD21.**


Despite recognising the benefits of deprescribing, many physicians reported significant gaps in their formal education regarding when and how to discontinue medications safely. The lack of structured deprescribing training during medical school and residency leaves some physicians uncertain about the best approach to deprescribing in clinical practice. Without clear institutional support or standardised deprescribing protocols, physicians may hesitate to deprescribe due to concerns about patient reactions, liability, or potential adverse effects. Additionally, limited access to deprescribing guidelines and decision–support tools can make the process more challenging, particularly in busy clinical settings were time constraints impact decision making.

“There are still gaps in deprescribing education. We need more structured training on when and how to stop medications safely.” 
**PD14.**


“During my medical training, deprescribing was rarely discussed. Most of our focus was on diagnosing and prescribing, but we were never really taught how to reassess medications and decide when they are no longer needed.” 
**PD18.**


Access to guidelines supported informed decisions, but inconsistent training across settings led to varied confidence in deprescribing.

A secondary review of participant responses suggested informal variation in deprescribing perspectives across medical specialities. For example, neurologists often raised concerns about withdrawal effects, while paediatricians frequently highlighted caregiver resistance. These illustrative differences are summarised in Appendix A.

## 4. Discussion

### 4.1. Main Findings

This study identified key behavioural determinants influencing proactive deprescribing among physicians in Saudi Arabian hospitals using the Theoretical Domains Framework (TDF). The primary domains affecting deprescribing practices were Environmental Context and Resources, Social Influences, Beliefs About Capabilities, Social/Professional Role and Identity, Skills, Beliefs about Consequences, and Knowledge. Physicians cited the availability of deprescribing guidelines, decision–support tools, and institutional policies as key facilitators. In contrast, high patient volume, limited time, and lack of structured deprescribing protocols emerged as significant barriers. Collaboration with pharmacists and geriatricians enhanced deprescribing efforts, but resistance from patients and caregivers posed challenges. Additionally, physicians with training in deprescribing demonstrated greater confidence, while knowledge gaps and concerns about legal and clinical consequences hindered deprescribing engagement.

### 4.2. Main Discussion

The findings highlight the complexity of deprescribing in hospitals, where structural and behavioural factors significantly influence physicians’ engagement in medication review. Environmental Context and Resources were central, with physicians emphasising the importance of deprescribing tools, institutional policies, and adequate consultation time. Research supports structured deprescribing interventions and clinical decision–support tools to improve physician confidence [30]. However, high patient volumes and outdated electronic health records (EHRs) were identified as significant barriers, consistent with studies indicating that workflow constraints impede deprescribing in acute care settings [31]. Unlike long-term care facilities, where deprescribing is integrated into routine medication reviews, hospitals often prioritise acute treatment, which limits opportunities for deprescribing [32]. Social Influences played a critical role, as collaboration with multidisciplinary teams facilitated deprescribing. Similar findings in prior studies emphasise the role of interprofessional collaboration in successful deprescribing [8]. However, resistance from patients and caregivers was a significant barrier, consistent with research highlighting that attachment to medications and fear of withdrawal symptoms reduce deprescribing acceptance [33]. Effective communication strategies, including shared decision making and patient education, are crucial in addressing deprescribing resistance and improving adherence to medication changes [34]. Physicians’ Beliefs about Capabilities and Skills were essential determinants of deprescribing behaviour. Those with specialised training in polypharmacy management reported greater confidence, whereas those lacking formal deprescribing education hesitated due to uncertainty about withdrawal effects and patient safety. These findings align with studies suggesting that insufficient deprescribing training contributes to physician reluctance [35]. Unlike long-term care settings, where deprescribing is integrated into routine practice, hospital environments often prioritise immediate treatment needs, limiting deprescribing opportunities [36]. Moreover, Beliefs about Consequences influenced physicians’ engagement with deprescribing. Those who observed positive patient outcomes following deprescribing were more likely to integrate it into their practice, aligning with previous research linking positive reinforcement with greater deprescribing confidence [37]. However, concerns about withdrawal effects, legal accountability, and potential harm discouraged deprescribing, reflecting studies indicating that defensive prescribing remains a key barrier in hospitals [38]. Social/Professional Roles and Identities further shaped physicians’ deprescribing behaviour. Physicians who viewed deprescribing as part of their duty to provide patient-centred care were likelier to engage in it. However, in settings where deprescribing was not emphasised institutionally, some physicians expressed uncertainty about whether it aligned with their professional role. In contrast, structured deprescribing policies in long-term care facilities encourage routine medication reviews, suggesting that hospitals may benefit from similar institutional support [39]. A critical barrier was the lack of Knowledge and formal deprescribing education. Many physicians reported that their training focussed primarily on prescribing rather than safe medication discontinuation. This aligns with literature indicating that structured deprescribing education improves physician confidence and willingness to deprescribe [40]. Expanding deprescribing education in medical curricula and offering targeted professional development programs could help bridge this gap. Among the 14 TDF domains, 7 were most frequently and strongly represented across interviews, particularly Environmental Context and Resources, Beliefs About Capabilities, and Social Influences. These domains reflect key behavioural determinants influencing deprescribing engagement and should be prioritised when designing interventions. Focussing on these areas may enhance the feasibility and impact of implementation strategies aimed at promoting proactive deprescribing in hospital settings. Although the analysis was not designed to compare across specialities, informal patterns emerged suggesting that clinical context may shape deprescribing attitudes. For instance, withdrawal concerns were more prominent among neurologists, whereas caregiver influence was a recurring issue in paediatrics. These insights, detailed in Appendix A, highlight the need for future research exploring speciality-specific barriers and tailored interventions. To translate the findings into actionable strategies, several targeted interventions aligned with the key TDF domains are proposed. First, under the domain of Environmental Context and Resources, piloting electronic health record (EHR)-integrated deprescribing alerts in a single hospital ward could prompt physicians during routine medication reviews by flagging potentially inappropriate medications and offering embedded access to relevant guidelines. Second, to address barriers related to Skills and Beliefs About Capabilities, interactive workshops focussed on deprescribing decision making and communication with patients and caregivers could be offered, particularly targeting early-career physicians. These sessions would help build confidence in managing complex cases. Third, aligned with the Social Influences domain, establishing multidisciplinary deprescribing rounds involving pharmacists, geriatricians, and medical teams could foster shared responsibility and peer learning. Together, these interventions represent practical steps to address behavioural determinants and support the integration of proactive deprescribing into routine hospital practice. In addition to multidisciplinary collaboration, integrating pharmacist prescribers into clinical teams may further optimise proactive deprescribing. Pharmacist prescribers have demonstrated effectiveness in medication management and deprescribing, particularly in reducing inappropriate prescriptions and enhancing patient outcomes [41].

### 4.3. Strengths and Limitations

This study has several strengths. First, it applied the Theoretical Domains Framework (TDF) to systematically explore the behavioural determinants of proactive deprescribing, providing a comprehensive understanding of the barriers and facilitators influencing physicians’ practices. Second, the study included a diverse sample of physicians across multiple specialities and regions in Saudi Arabia, enhancing the generalisability of the findings within hospital settings. Third, the qualitative approach enabled an in-depth exploration of physicians’ perspectives, uncovering nuanced barriers and enablers that quantitative methods might not capture. However, some limitations must be acknowledged. First, the study focussed exclusively on hospital settings, limiting its applicability to other healthcare environments, such as primary care or long-term care facilities, where deprescribing dynamics may differ. Second, while efforts were made to include a diverse sample, the predominance of senior physicians in the study may not fully represent the perspectives of early-career physicians, who may face different deprescribing challenges. Transcripts were not returned to participants for member checking, which may limit credibility; however, this was mitigated through reflexive journaling and peer debriefing. Additionally, while the findings offer insights into hospital settings, their transferability to primary care or long-term care may be limited due to differences in care continuity, prescribing autonomy, and organisational structure. An additional limitation is the potential influence of the interviewer and cultural norms on participants’ responses, which may have shaped how openly views were expressed. Finally, the study relied on self-reported data, which may be subject to recall or social desirability bias.

### 4.4. Implications

The findings carry significant implications for clinical practice and policy. To address key environmental barriers such as time constraints, lack of standardised protocols, and limited clinical tools, hospitals should integrate deprescribing alerts into electronic health records, implement evidence-based guidelines, and embed deprescribing into routine workflows. To overcome knowledge gaps and low confidence among clinicians (Beliefs About Capabilities and Knowledge), structured deprescribing training should be incorporated into undergraduate curricula and continuing professional development. Patient and caregiver resistance (Social Influences) can be mitigated through shared decision-making approaches and communication training that supports physician–patient dialogue. These strategies are particularly relevant to Saudi Arabia’s healthcare transformation under Vision 2030. Embedding proactive deprescribing into national clinical guidelines and CBAHI accreditation standards, such as through mandated multidisciplinary medication reviews, may improve medication safety, reduce avoidable healthcare costs, and promote rational prescribing aligned with value-based care goals.

## 5. Conclusions

This study highlights the multifaceted barriers and facilitators influencing proactive deprescribing in hospital settings, particularly within the domains of Environmental Context and Resources, Social Influences, and Beliefs About Capabilities. Targeted interventions such as EHR-integrated deprescribing alerts to address system-level barriers, structured clinician training to build confidence and knowledge, and multidisciplinary collaboration to enhance team-based care can directly address these challenges. While grounded in the Saudi context, the findings apply to other healthcare systems facing similar pressures from ageing populations and polypharmacy. Advancing a culture of proactive deprescribing requires system-level commitment, clinician engagement, and policy support to embed safe medication optimisation into routine hospital practice. This research contributes to the broader effort to promote patient-centred care, reduce medication-related harm, and establish deprescribing as a global priority in clinical quality and safety.

## Figures and Tables

**Table 1 healthcare-13-01274-t001:** Demographic and professional characteristics of participating physicians.

Participant Identifier	Age Range (Years)	Gender	Years of Experience	Specialty	City of Practice	Region of Practice
PD1	45–54	Female	11–20 years	Physical medicine and rehabilitation	Jeddah	Western Region
PD2	55–64	Male	More than 20 years	Paediatrics		
PD3	55–64	Male	More than 20 years	Orthopaedic surgeon		
PD4	55–64	Male	More than 20 years	Internal medicine	Makkah	
PD5	55–64	Male	More than 20 years	Paediatrics		
PD6	35–44	Male	11–20 years	Neurology		
PD7	45–54	Female	11–20 years	Physical medicine and rehabilitation		
PD8	55–64	Male	More than 20 years	Oncology	Taif	
PD9	55–64	Male	More than 20 years	Oncology		
PD10	35–44	Male	11–20 years	Neurology	Al-Kharj	Central Region
PD11	55–64	Male	More than 20 years	Paediatrics		
PD12	35–44	Male	11–20 years	Neurology		
PD13	45–54	Female	11–20 years	Physical medicine and rehabilitation		
PD14	55–64	Male	More than 20 years	Internal medicine		
PD15	35–44	Male	11–20 years	Neurology	Riyadh	
PD16	25–34	Male	Less than 5 years	General Practitioner		
PD17	55–64	Male	More than 20 years	Paediatrics		
PD18	55–64	Male	More than 20 years	Oncology	Buraydah	
PD19	25–34	Male	Less than 5 years	General Practitioner		
PD20	55–64	Male	More than 20 years	Oncology	Abha	Southern Region
PD21	55–64	Male	More than 20 years	Paediatrics		
PD22	55–64	Male	More than 20 years	Paediatrics	Jazan	
PD23	45–54	Female	11–20 years	Physical medicine and rehabilitation	Dammam	Eastern Region
PD24	25–34	Male	Less than 5 years	General Practitioner		
PD25	35–44	Male	11–20 years	Neurology		
PD26	35–44	Male	11–20 years	Neurology	Al-Khobar	
PD27	45–54	Female	11–20 years	Physical medicine and rehabilitation	Jubail	

## Data Availability

The data are available upon request from the corresponding author.

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
