# Peer review of "Barriers and Facilitators to Proactive Deprescribing in Saudi Hospitals: A Qualitative Study Using the Theoretical Domains Framework"

_healthcare, 2025, doi:10.3390/healthcare13111274_

Round 1

Reviewer 1 Report

Comments and Suggestions for Authors

This manuscript tackles a timely and important issue—understanding what helps and hinders physicians in Saudi hospitals from proactively deprescribing medications. The use of the Theoretical Domains Framework (TDF) lends a strong theoretical underpinning, and the qualitative approach offers rich, nuanced insights. Overall, the study has merit and could make a valuable contribution to efforts aimed at improving medication safety. However, several areas would benefit from deeper methodological transparency, stronger linkage to local practice contexts, and clearer articulation of how findings can drive concrete interventions.

#Introduction & Rationale
(1) The author clearly distinguish reactive from proactive deprescribing. Can your strengthen the paper to ground this distinction in concrete examples from Saudi practice ?

(2) Can the author elaborate on why physicians in Saudi Arabia in particular may face unique cultural or system-level barriers (e.g., prescribing norms, medico-legal climate, role of extended family in care decisions)?

#Methods
(a) Sampling & Recruitment

(1) The purposive sampling frame is described broadly, but it isn’t clear how you ensured representation across key axes (junior vs. senior, public vs. private, urban vs. rural hospitals). Given that 48% of respondents have >20 years’ experience, how might early-career perspectives differ?

(2) You note no minimum experience requirement, yet the final sample skewed senior. Was this by design or necessity? Please clarify how you determined saturation—did you stop because no new TDF themes appeared, or due to logistical constraints?

(b) Interview Guide & Data Collection

(1)The manuscript states the guide was developed via literature review and expert consultation. Would the author share a brief overview of key pilot questions or an example excerpt? This helps readers assess how comprehensively you probed each TDF domain.

(2) Who did conduct the interviews, and what steps were taken to minimise interviewer bias? For instance, was there any reflexive journaling or peer debriefing to surface assumptions?

(c) Coding & Analysis
(1)Manual coding into 14 TDF domains is appropriate, but did you employ any software (e.g., NVivo) to manage data? Was coding cross-checked by a second researcher (which might be problematic because there is only one author listed on this paper) to enhance reliability? If not, please discuss how you maintained consistency.

(2)Were any themes or insights identified that fell outside the predefined TDF domains? If so, how did you handle them?

#Results Presentation
(1) Table 1 is helpful for demographic context, but consider improve it by adding columns for “Hospital Type” (teaching vs. community) or “Ward Setting” (e.g., ICU, general medicine).

(2) In each domain subsection, you provide illustrative quotes—excellent. To demonstrate depth, you might note variations (e.g., did neurology vs. pediatrics physicians express differing concerns about withdrawal effects?).

#Discussion & Practical Implications
(1) You map findings to existing literature nicely. To move from “what is” to “what to do,” please propose 2–3 targeted interventions tied to TDF domains. For example, under Environmental Context, recommend a pilot of EHR-embedded deprescribing alerts in one ward and measure uptake.

(2) Given Saudi Arabia’s push toward value-based care, can you consider discussing how your findings could inform national policy or accreditation standards around medication review?

#Limitations
(1)You note senior-physician predominance and reliance on self-report. Please also discuss the absence of member-checking (if transcripts were not returned to participants), as this affects credibility.

(2) Reflect on transferability: how might your findings apply (or not) to primary-care or long-term care settings in the Kingdom?

#Conclusions & Next Steps
The conclusion restates key barriers/facilitators. Strengthen this by calling out immediate next steps—e.g., co-designing a deprescribing toolkit with frontline clinicians or running a small feasibility study.

#Formatting & Consistency
(1) In Funding you first state “no financial support” but in Acknowledgments mention funding by Taif University’s Deanship. Please reconcile these statements.

(2) A few minor typos: “were not directly involved in medication management.” (Methods, line 114) – perhaps “not directly involved in medication adjustment”? And in Table 1 legend, define acronyms (e.g., “SD” if used).

Comments on the Quality of English Language

The English level is good

Author Response

Dear Reviewer, 

Please find the author's responses to the reviewers' comments. The revised copy is also attached in the files. 

Regards

Reviewer 2 Report

Comments and Suggestions for Authors

The author conducts a qualitative study aimed at the identification of barriers and facilitators to implementation of proactive deprescribing among physicians in Saudi Arabian Hospitals. The methodological approach is based on application of the Theoretical domains Framework. Recorded semi-structured interviews were manually analyzed and coded according to the 14 TDF domains. The analysis is not focused on a specific drug class or on a specific medical area of intervention.

Despite the limitation declared by the author of providing a partial point of view due to sample recruitment in hospital setting and involving only physicians and not other healthcare professionals, the results of the study are easily generalizable and obtained through a methodologically correct approach.

Nevertheless, the author should make an effort to scale up towards a progression from investigation to intervention.

The aim of the application of the Theoretical Domains Framework could be the identification of a list of relevant theoretical constructs on the base, for example, of criteria such as the relatively high frequency of their detection in the interviews.  These themes are those which most likely influence implementation and represent the problems to prioritize in searching solutions for implementation. We suggest to the authors to revise the Discussion section according to this perspective.

Another suggestion to the author is to review the study in accordance with the Consolidated criteria for Reporting Qualitative research (COREQ). The guidance of the checklist will surely help to ameliorate important aspects of the paper, such as the study methods, context of the study, findings, analysis and interpretations.

As a conclusion, I recommend the paper publication with minor revision in order to add to the paper innovative elements of analysis.

Author Response

(The authors gave the same response as above.)

Reviewer 3 Report

Comments and Suggestions for Authors

Barriers and Facilitators to Proactive Deprescribing Among  Physicians in Saudi Hospitals: A Qualitative Study Using the 3 Theoretical Domains Framework

REVIEW

The authors have written about an understudied and clinically relevant topic. This topic is highly relevant to this journal. The purpose of the research is very well-defined, and I'm sure the objectives will be met. Generally, the paper has high issues with the writing standard, and the tables are not manipulated.

Despite these positive issues, the paper has some important limitations that should be discussed. The Methods should be expanded, and some other limitations (e.g., selection criteria), methodological flows, and some interventions should be better described.

However, the current form presented requires revision before consideration for publication.

The manuscript could be strengthened by attending to the following matters:

GENERAL COMMENTS:

Positive:

  • Important topic

  • Novelties

  • Clinical topic

Negative:

  • Some questions in Methods

Abstract

Maybe the abstract is too long. I will cut some sentences.

Not clear how the author identified key factors.

Introduction

I think that the authors could mention the prevalence of polypharmacy. Defining polypharmacy in older adults: a cross-sectional comparison of prevalence estimates calculated according to active ingredient and unique product counts. Int J Clin Pharm (2025). https://doi.org/10.1007/s11096-025-01882-7

The authors could mention that collaboration with clinical pharmacists significantly reduced polypharmacy and inappropriate prescribing and errors on the transition of care: Clinical pharmacist interventions in the transition of care in a mental health hospital: case reports focused on the medication reconciliation process. 

Please put the aim of the study in a separate paragraph. Please focus on the main aim of the study.

Methods

Not clear how the author selected appropriate physicians.

Why did the author select this sampling?

The author did not mention trustworthiness.

Did the authors predefine questions, or how did they select questions?

Why did the author conduct the interviews for 30-60 minutes? If the questions were not standardised, there would have been many different questions.

How did the author transform the interview into the codes?

Results

Please also include absolute numbers following to the percentages.

How did the author identify seven domains (in line with the framework)? Did the authors selected 7 from 14 domains?

Environmental Context and Resources

First paragraph: General statements are given without appropriate references.

Social Influences

First paragraph: General statements are given without appropriate references.

Discussion

The authors should mention that one of the possible approach to optimize this field is adding pharmacists prescribers to the team.

Author Response

(The authors gave the same response as above.)

Reviewer 4 Report

Comments and Suggestions for Authors

As per the attached PDF review file. 

Author Response

(The authors gave the same response as above.)
